# Effects of Particle Size on Physicochemical and Nutritional Properties and Antioxidant Activity of Apple and Carrot Pomaces

**DOI:** 10.3390/foods13050710

**Published:** 2024-02-26

**Authors:** Saeed Salari, Joana Ferreira, Ana Lima, Isabel Sousa

**Affiliations:** 1LEAF—Linking Landscape, Environment, Agriculture and Food—Research Centre, Instituto Superior de Agronomia, Universidade de Lisboa, Tapada da Ajuda, 1349-017 Lisboa, Portugal; saeedsalari@isa.ulisboa.pt (S.S.); isabelsousa@isa.ulisboa.pt (I.S.); 2Veterinary and Animal Research Centre (CECAV), Faculty of Veterinary Medicine, Lusófona University, 376 Campo Grande, 1749-024 Lisbon, Portugal; agusmaolima@gmail.com

**Keywords:** apple pomace flour, carrot pomace flour, waste valorization, nutritional composition, particle size

## Abstract

The food processing industry is growing rapidly and producing large amounts of by-products, such as pomaces, which are considered as no-value waste and cause significant environmental pollution. The main by-products of fruit juice processing companies are apple and carrot pomaces, which can be used to create new functional food products. In the present study, the effects of particle size (PS) on the proximate composition, nutritional properties, and antioxidant activity of apple pomace flour (APF) and carrot pomace flour (CPF) were determined. Four different PS fractions, PS > 1 mm, 1 > PS > 0.71 mm, 0.71 > PS > 0.18 mm, and 0.18 > PS > 0.075 mm were used for the present study. Their vitamin, carotenoid, organic acid, and reducing sugar contents were determined using HPLC. The proximate compositions of each PS fraction of the AP and CP flours were determined using recommended international standard methods. DPPH, FRAP, and Folin-Ciocalteu methods were used to measure their antioxidant activity and total phenolic compounds, respectively. The moisture content (around 12.1 mg/100 g) was similar in all PS fractions and in both flours. The APF had lower protein (4.3–4.6 g/100 g dw) and ash (1.7–2.0 g/100 g dw) contents compared to the CPF, with protein contents ranging from 6.4–6.8 g/100 g dw and ash contents ranging from 5.8–6.1 g/100 g dw. Smaller particles, regardless of flour type, exhibited higher sugar and phenolic contents and antioxidant activity, while vitamins were more abundant in particles larger than 1 mm. In the APF, larger particles had a higher fiber content than smaller particles, while their fat content was the lowest. PS also had an impact on the results of the carotenoid contents. This study underscores the direct impact of PS on the distribution of sugars, crude fiber, fat, carotenoids, vitamins, total phenolic compounds, and antioxidant activity in pomaces.

## 1. Introduction

As societal awareness increases regarding the impact of food on human health and the environment, a notable shift has been unfolding, encompassing a widespread inclination of consumers toward clean-label foods with natural health-promoting ingredients whilst also promoting sustainability and reducing waste [1,2,3]. Fruit pomaces, the often-overlooked by-products of fruit juice companies, are one of the best candidates to fit this shift, since they are particularly rich in bioactive compounds and essential nutrients and can be a good source of flavor and color. However, they often constitute a disposal problem. During juice production, 30 to 50% of the original apples and carrots are wasted as pomaces [4,5,6,7]. Due to the abundance of pomaces generated by juice industries, their use would be cost-effective, highly sustainable, and a promoter of the circular economy. Thus, this possibility has sparked a surge of research interest to unveil the untapped potential of upcycling pomaces as natural ingredients. This holds promise for developing novel functional foods, offering a positive impact on human health while using natural ingredients and reducing the need for preservers and other synthetic chemicals [5,8].

The bioactive compounds in these pomaces refer to a wide variety of molecules with different chemical structures such as polyphenols, carotenoids, vitamins, omega-3 fatty acids, organic acids, nucleosides, nucleotides, and phytosterols [9]. Aside from their nutritional properties, these molecules may have preservative and health-promoting potential through antibacterial, anti-inflammatory, and antioxidant activities [10,11,12,13]. Within this context, fruit pomaces can be of tremendous potential for consumers and industry alike. Their use as food ingredients is an efficient way to introduce flavor, nutrients, and important bioactivities in foods. Moreover, various studies have consistently revealed a correlation between the consumption of bioactive-rich foods and a reduced risk of various chronic diseases, including cancer, cardiovascular diseases, Alzheimer’s, and diabetes [14,15,16,17,18].

Several factors strongly affect the functionality of pomaces, such as drying processes and particle size (PS) [19,20,21]. To the best of our knowledge, there is a lack information in the literature related to the impact of PS on the nutritional composition and biological activity of apple and carrot pomaces. Hence, the evaluation of the distribution of these features in different pomaces could be significant to improve and optimize the sustainable use of these types of pomaces as food ingredients to enhance the nutritional status of foods or to promote the development of functional foods with added benefits for consumer health.

In this study, we aimed to investigate the impact of different PS fractions comparatively and extensively on nutritional composition, including sugar, fiber, vitamins, phenolic compounds, and carotenoids, as well as the antioxidant activity of pomace flours from fruit and vegetable sources. This type of approach can make a substantial contribution to the valorization of these byproducts within the food industry, ultimately leading to a reduction in their environmental impact.

## 2. Materials and Methods

### 2.1. Chemicals

All the reagents used were of analytical or HPLC grade. Methanol and ethanol (96%) were purchased from Fisher Chemicals (Fair Lawn, NJ, USA). DPPH (2,2-diphenyl-1-picrylhydrazyl), Trolox (6-hydroxy-2,5,7,8-tetramethylchromane-2-carboxylic acid), acetic acid, sodium acetate, TPTZ (2,4,6-tris(2-pyridyl)-s-triazine), hydrochloric acid, iron (III) chloride hexahydrate, Folin-Ciocalteau reagent, sodium carbonate, gallic acid, MES monohydrate, tris(hydroxymethyl)aminomethane, L(+)-ascorbic acid, DL-malic acid, L(+) tartaric acid, β-carotene, vitamin B6, and vitamin E were purchased from Sigma-Aldrich (St. Louis, MO, USA). D(+)-Glucose and D(−)-Fructose were purchased from VWR International (Radnor, PA, USA). Lutein was purchased from Thermo Fisher Scientific (Waltham, MA, USA). Celite was purchased from AppliChem (Darmstadt, Germany). Sodium hydroxide and Acetone were purchased from LabChem (Zelienople, PA, USA). Ultrapure water was obtained from a Synergy^®^ Water Purification System (MilliporeSigma, Burlington, MA, USA).

### 2.2. Plant Material and Sample Preparation

APF and CPF, obtained from a juice processing company in Portugal (ALITEC—ALIMENTOS TECNOLÓGICOS, S.A., Nazaré, Portugal), were submitted to a process of tunnel-drying (Tecnofruta, Valencia, Spain) under certain conditions (80–85 °C, 110 min, 55 Hz of air flow) and grinding (Ferneto, Vagos, Portugal). The APF samples were from Gala, Fuji, and Granny varieties, and the CPF samples were from Berlicum, Nantes, and Kuroda varieties. The samples were fractionated based on PS using sieves with the following mesh sizes: 1, 0.71, 0.18, and 0.075 mm. Henceforth, the fractions are denoted by uppercase Latin letters as follows: A = original whole pomaces, B = PS ≥ 1 mm, C = 1 mm > PS ≥ 0.71 mm, D = 0.71 mm > PS ≥ 0.18 mm, and E = 0.18 mm > PS ≥ 0.075 mm.

### 2.3. Proximate Composition Analysis

A proximate composition analysis of the APF and CPF fractions was carried out following different international standard methods. Each analysis was carried out in triplicate.

In brief, the moisture content of each sample was determined according to the standard method AACC 44-15.02 [22]: 2 g of each sample was weighed (Denver Instrument Company, TC-403) and placed in a stove (BINDER, ED56, Tuttlingen, Germany) at 105 ± 1 °C until it was a constant weight. The total ash content was measured via incineration according to the standard method AACC 08-01.01 [23] using a muffle (SNOL, Utena, Lithuania) at 550 ± 1 °C until the samples were a constant weight. The ash was cooled in a desiccator and weighed using a digital scale (Denver Instrument Company, TC-403, Denver, CO, USA). The mineral contents (Na, K, Ca, Mg, P, S, Fe, Cu, Zn, Mn, and B) were estimated using inductively coupled plasma optical-emission spectrometry (iCap Series-7000 plus series ICP-OES, Thermo Fisher Scientific, Waltham, MA, USA) following the method described by AACC 40-75.01 [24].

The protein content of the samples was estimated using a combustion method following ISO 16634-2:2016 [25] using DUMAS equipment (VELP SCIENTIFICA, NDA 702, Usmate, Italy) and a conversion factor of 6.25. The samples were combusted at a temperature ranging from 800–1000 °C, converting all nitrogen forms to nitrogen oxides. These nitrogen oxides were then reduced to nitrogen gas (N_2_) and measured using a thermal conductivity detector.

The total fat of the samples was determined following the standard method AACC 30-25.01 using the Soxhlet method with hexane for 6 h. The extracted lipids were concentrated using a rotary evaporator, dried in an oven, and weighed. The total dietary fiber (TDF) content was evaluated using the enzymatic–gravimetric method (AOAC method 991.43) [26]. The pH of the samples was measured based on a suspension obtained by mixing each sample (1 g) with 5 mL of deionized water using a pH meter (Crison Instruments, S.A., Barcelona, Spain).

### 2.4. Extract Preparation

To 2 g of each pomace flour sample, 20 mL of ethanol (96%) was added (1:10 *v*/*v*) and mixed for 5 h using an overhead shaker (Heidolph Instruments, Schwabach, Germany) at room temperature. Subsequently, the samples were centrifuged at 3220× *g* for 10 min. The supernatants were recovered and stored at −24 °C.

### 2.5. Sugar and Organic Acid Composition

The sample extracts (1:20 *v*/*v*) were diluted in 25 mM sulfuric acid. Separation of the organic acids and sugars from all the extracts was accomplished using a high-performance liquid chromatography (HPLC) system (Chromaster, Hitachi, Tokyo, Japan) equipped with a UV–Vis detector 5420 (Chromaster, Hitachi, Tokyo, Japan) monitored at 210 nm for the organic acids and a Refractive Index (RI) Detector 5450 (Chromaster, Hitachi, Tokyo, Japan) for the sugars. Separation of the organic acids and sugars was achieved using an ionic exclusion column (Rezex™ ROA Organic Acid H+ (8%) column, 300 × 7.8 mm, Phenomenex, Torrance, CA, USA) at 65 °C. Sulfuric acid (5 mM) was used as a mobile phase at a flow rate of 0.5 mL/min. Organic acids identification and quantification was achieved using malic acetic, ascorbic, and tartaric acids as standards, with concentrations ranging from 0 to 20 g/L. The results were represented in g/L of the corresponding organic acid (calibration curves: y = 4.48 × 10^−7^, R^2^ = 0.9998 for malic acid; y = 7.31 × 10^−7^, R^2^ = 0.9996 for acetic acid; y = 7.83 × 10^−7^, R^2^ = 0.9998 for lactic acid; y = 4.12 × 10^−7^, R^2^ = 0.9994 for citric acid; y = 2.39 × 10^−7^, R^2^ = 0.9997 for tartaric acid; and y = 5.37 × 10^−7^, R^2^ = 0.9998 for ascorbic acid). The experiments were performed in triplicate. Sugar identification and quantification was achieved using glucose and fructose as standards, with concentrations ranging from 0 to 40 g/L. The results were expressed in g/L of the corresponding sugar (calibration curves: y = 1.48 × 10^−6^, R^2^ = 0.9999 for glucose and y = 1.42 × 10^−6^, R^2^ = 0.9997 for fructose). The experiments were performed in triplicate. Chromquest version 4.2 software was used for data interpretation.

### 2.6. Carotenoid and Vitamin Compositions

Separation of the carotenoids and vitamins from all the extracts was accomplished using a high-performance liquid chromatography (HPLC) system (Dionex^TM^, Thermo Fisher Scientific, CA, USA) equipped with a Diode Array Detector UltiMate 3000 (Dionex, Thermo Fischer Scientific, CA, USA) monitored at 450 nm for carotenoids and 290 and 220 nm for vitamins. Separation of the carotenoids was achieved using a YMC Carotenoid C30 column (250 × 4.6 mm, 3 μm, YMC America, Devens, MA, USA) at ambient temperature. Two mobile phases of A (MeOH/MTBE/H_2_O = 81/15/4) and B (MeOH/MTBE/H_2_O = 6/90/4) were used at elution gradient of 1–100% B for 90 min and at a flow rate of 1 mL/min. Separation of the vitamins was achieved using a XTerra MS C18 column (250 × 4.6 mm, 5 μm, Waters Limited, Wilmslow, Cheshire, England) at 40 °C. Phosphate buffer (10 mM TBAOH and 10 mM KH_2_PO_4_) with a pH of 5.2 and acetonitrile with the ratio of 90:10 *v/v* were used as mobile phases for the water-soluble vitamins at a flow rate of 0.8 mL/min. For the fat-soluble vitamins, acetonitrile and MeOH with a ratio of 60:40 *v/v* were used at a flow rate of 1 mL/min. Dionex Chromeleon version 6.8 software was used for data interpretation.

### 2.7. Total Phenolic Content (TPC) and Antioxidant Activity (AAT)

As previously suggested by ref. [27], the total phenolic content was ascertained by applying the Folin Ciocalteu technique using gallic acid as the standard. A total of 140 μL of Folin Ciocalteu reagent (Sigma-Aldrich, St. Louis, MO, USA) and 2.4 mL of deionized water were added to 150 µL aliquots of each sample, and the mixture was vortexed. After 3 min, 300 µL of sodium carbonate was added, vortexed once more, and left to rest at room temperature for two hours in the dark. At 725 nm, the absorbance of every sample was determined. The results were represented in mg GAE per 100 g of sample dw (gallic acid calibration curve: 0 to 200 μg·mL^−1^, R^2^ = 0.9998). The experiments were performed in triplicate.

The antioxidant capacity was assessed using DPPH and FRAP assays as follows. The DPPH experiment was conducted by combining 100 µL of sample extract with 3.9 mL of DPPH radical solution (0.06 mM in methanol) to assess the radical scavenger capacity. The reaction mixture was placed in a 15 mL plastic tube which was screwed closed, vortexed for 30 s (RSLAB-6PRO, Normax, Marinha Grande, Portugal), and incubated for 40 min in a circulating bath (Precision^TM^ Thermo Scientific, Waltham, MA, USA) at room temperature, protected from light. After this time, the absorbance was measured using a UV–visible spectrophotometer (Cary 100, Agilent Technologies, Santa Clara, CA, USA) using a 1 cm thick quartz cuvette at 515 nm against methanol. For the blank sample, 0.1 mL of methanol was added instead of the extract. The standard curve was prepared based on a methanolic solution of Trolox (6-hydroxy-2,5,7,8-tetramethylchroman-2-carboxylic acid) with concentrations ranging from 0 to 1000 µg·mL^−1^, R^2^ = 0.9958. The samples were analyzed in triplicate, and the results were expressed as mg of Trolox per 100 g dry weight (dw).

A FRAP assay was also used to assess the antioxidant capacity. The analysis was performed according to the procedure previously described [28] with modifications. FRAP (ferric reducing antioxidant power) test solution was prepared by mixing at a ratio of 10:1:1 (*v*/*v*/*v*) 0.3 M acetate buffer at pH 3.6, 10 mM TPTZ (2,4,6-Tris(2-pyridyl)-s-triazine) prepared in 40 mM HCl solution, and 20 mM aqueous iron (III) chloride hexahydrate solution. A total of 90 µL of sample extract, 270 µL of distilled water, and 2.7 mL of test solution were placed in a 15 mL plastic test tube which was screwed closed, vortexed for 30 s (RSLAB-6PRO, Normax, Marinha Grande, Portugal), and incubated for 30 min in a water bath (Precision™ 2864, Thermo Scientific, Waltham, MA, USA) at a temperature of 37 °C, protected from light. After this time, the absorbance was measured using a UV–visible spectrophotometer (Cary 100, Agilent Technologies, Santa Clara, CA, USA) using a 1 cm thick quartz cuvette at a wavelength of 595 nm against the test solution. The standard curve was prepared based on a methanolic solution of Trolox with concentrations ranging from 0 to 800 µg.mL^−1^, R^2^ = 0.9971. The samples were analyzed in triplicate, and the results were expressed as mg of Trolox per 100 g dw.

### 2.8. Statistical Analysis

All analyses were performed in triplicate, and the data are expressed as means ± standard deviations (SDs). The significance between different fractions was evaluated using a one-way ANOVA using SPSS software (version 24, IBM, Endicott, NY, USA) following by a Tukey post hoc test at a significance level of 95% (*p* < 0.05).

## 3. Results and Discussion

### 3.1. Particle Size Distribution

The APF and CPF samples were sieved through consecutive sieves with the following mesh sizes: 1, 0.71, 0.18, and 0.075 mm. After sieving, the mass retained in each sieve was weighed, and the corresponding mass fraction yields were determined. The results are summarized in Figure 1.

The primary composition of the APF and CPF largely consisted of particles ranging between 0.18 mm and 0.71 mm (referred to as fraction D), accounting for 44.2% to 68.7% of the overall bulk sample. Specifically, within the APF, the least-prevalent fraction (9.7%) comprised smaller particles ranging from 0.18 to 0.075 mm (designated as fraction E). Conversely, in the CPF, the fraction containing particles equal to or larger than 1 mm (referred to as fraction B) exhibited the lowest proportion (4.1%). The disparate distributions of PS in the APF and CPF stem primarily from their distinct structural and anatomical characteristics. In the case of the APF, components like the lignified portions encompassing the stalk, skin, and seeds offered resistance during the grinding process post-dehydration, resulting in larger particles. Conversely, the CPF, largely composed of pulp, lacked substantial resistance during grinding, yielding particles of smaller sizes.

### 3.2. Proximate Composition

APF and CPF can be an economically important sources of protein, fat, fiber, carbohydrates, vitamins, and minerals in the diets of many individuals in developed and developing countries. The proximate composition of these food provides information on their common nutrients.

#### 3.2.1. Moisture Content

The moisture content for the different fractions of APF and CPF ranged between 11.2 and 13.0 g/100 g and 11.1 and 12.9 g/g, respectively (Figure 2).

In the APF, only the fractions with a PS of more than 1 mm and above 0.075 mm showed significantly different moisture contents. With that being said, the fraction with particles of 1 mm (A) had the lowest moisture content (11.2 g/100 g), and the opposite was observed for the finer PS (fraction E), with a moisture content of approximately 13.0 g/100 g. The same trend was observed for the CPF, where the fractions with small-sized particles (fraction E) showed the highest moisture content (12.9 g/100 g). This was the opposite of what was first expected, since higher water absorption can be related to the presence of more fibrous materials that are mostly present in stalks, skins, and seeds (which mainly constitute the fractions of higher PS). However, the higher moisture absorption in the fractions of a smaller PS can be explained by the higher surface area of these particles, thereby absorbing more moisture from the atmosphere [29]. This information is important for the food processing industry, as moisture content directly influences the storage shelf life of pomaces.

#### 3.2.2. Protein Content

In the APF and CPF, there were no significant differences between the fractions, but there was a slight decrease from fraction A to E observed in the CPF. Fraction E had the lowest protein content: 4.3 and 6.4 g/100 g dw for APF and CPF, respectively (Figure 3).

A reduction in PS has been consistently associated with a corresponding decrease in protein content across different food matrices [30]. The average crude protein content ranged from 4.3–4.6 g/100 g dw (*p* < 0.05) and 6.4–6.8 g/100 g for the APF and CPF, respectively, which were almost in agreement with the data reported in the literature [4,31]. The higher protein content observed in the CPF can be due to the physiological nature of the carrots. As a plant root, carrots can absorb more nitrogen due to the availability of nitrogen in the soil and through fertilizers; therefore, they have a higher overall nitrogen content [32,33].

#### 3.2.3. Fat Content

The APF had a considerably higher fat content when compared to the CPF (Figure 4). This difference can be attributed to the different tissues that constitute these flours. APF contains seeds that are rich in fatty acids, particularly unsaturated fatty acids [34]. Also, apple peel has some waxy substances that are categorized in the fat fraction. Regarding the CPF, the particles larger than 1 mm had higher fat contents than the smaller particles (almost 2-fold); however, the amount of fat considering all the fractions was almost negligible. The total crude fat content in the APF fractions ranged between 0.5 and 1.5 g/100 g dw, and the highest fat content was observed in the fraction with a PS of 0.71 ≤ PS < 0.18 mm (fraction D). There seemed to be a decrease in the fat content of the fractions with a higher PS, which can be attributed to the lower surface area of that particles, decreasing the extraction efficiency, as documented in the literature [35]. Smaller particles allowed for more efficient breakdown of cell walls, releasing fats into the extracted solution.

#### 3.2.4. Total Dietary Fiber

TDF generally has low or no nutritional value. It measures the indigestible cellulose, pentosans, lignin, and other components. However, fiber is well known to reduce risk of diabetes, cardiovascular disease, and blood cholesterol [36,37]. TDF was analyzed in all the different PS fractions of the APF and CPF. Generally, it was observed that the APF had a higher total dietary fiber content (27.9–37.7 g/100 g dw) than the CPF (17.2–28.8 g/100 g dw) (Figure 5).

The results show a significant positive correlation between the PS of the APF and CPF and the TDF content. For both pomace flours, it was observed that the fractions with a smaller PS contained significantly lower contents of fiber (27.9–32.0 g/100 g dw and 17.2–21.7 g/100 g dw) compared to the coarser particle fractions, where the fiber content ranged between 34.9 and 37.7 g/100 g dw and 26.0 and 28.8 g/100 g dw in the APF and CPF, respectively. This decrease in crude fiber content can be due to two possible reasons: (1) the degradation of the main constituents of the pomace flours (hemicellulose, cellulose, and lignin), which are turned into small molecular substances after grinding [38]; (2) the extraction process can result in a higher release of soluble fibers; however, the overall measured fiber content may appear lower due to the breakdown of insoluble fiber structures during the extraction process [39].

#### 3.2.5. Ash Content

Determining the ash content in food samples is crucial for understanding their complete nutritional profile, quality, and microbiological stability. The ash content in food refers to the mineral and inorganic residue left behind after subjecting a sample to high temperatures, eliminating moisture, volatiles, and organics [40]. The predominant minerals and inorganics found in ash include calcium, magnesium, sodium, and potassium, with traces of manganese, zinc, and iron, among others, present in smaller amounts. Generally, food contains about 7% ash content, yet this figure can fluctuate. Ash content serves as an indicator of processing levels; natural foods typically possess lower ash contents compared to their highly processed counterparts. Pure oils and fats can register as low as 0% ash content, while processed dried meats may contain up to 12% ash [40].

The ash content of the APF and CPF ranged between 1.7 and 2.0 g/100 g dw and 5.8 and 6.1 g/100 g dw, respectively (Figure 6). Due to the nature of carrots as a root, the ash content of the CPF was significantly higher than the APF.

#### 3.2.6. Mineral Content

Minerals are essential nutrients with numerous health benefits. Sodium plays a crucial role in maintaining proper water balance within the body [41]. Potassium, on the other hand, has been associated with a reduced risk of stroke [42]. Calcium is essential for promoting the strength of bones and teeth [43], while magnesium has been linked to the prevention of depression [44]. Iron, vital for preventing anemia, has far-reaching implications, impacting physical work capacity, pregnancy outcomes, and the cognitive, motor, and behavioral development of children [45]. Additionally, zinc has been suggested to have positive effects on both the immune system and mental health [46,47].

According to Table 1, potassium was the most abundant mineral in both of the flours, which is in accordance with the literature [1,2], ranging from 872.8 to 925.3 mg/100 g dw and from 2269.9 to 2468.6 mg/100 g dw for the APF and CPF, respectively. Na, Ca, Mg, P, S, and Fe were the other key elements of both pomace samples, regardless the PS (Table 1). The predominance of these minerals in the APF and CPF has been previously reported [48,49], but the total mineral content was almost three-fold in the CPF (3564.1 mg/100 g dw) compared to the APF (1284.3 mg/100 g dw), considering the values for original whole pomaces (fraction A). According to the dietary reference values (DRV) for adults (≥18 years) presented in Table 1, both pomaces could be considered as excellent sources of essential minerals.

### 3.3. Carbohydrate and Organic Acid Contents

According to Figure 7, the APF had a markedly higher content of monosaccharides, consisting of glucose and fructose, when compared to the CPF. The underlying reason for this observation is correlated with the maturation of fruit, during which more energy is required for metabolic reactions [51]. The APF was particularly rich in fructose (around 20.68 g/100 g dw) compared to glucose (8.33 g/100 g dw). The same was true of the CPF, but to a lesser extent: 7.03 g/100 g dw of fructose and 6.24 g/100 g dw of glucose. As the PS of pomace flour decreased, the carbohydrate contents tended to rise. The smallest PS fractions (D and E) of the APF and CPF exhibited the highest levels of glucose and fructose, while the largest PS fraction (B) showed the lowest levels (see Figure 2). This correlation contributes to an escalation in sugars, which serve as a primary energy source in fruits. Additionally, the findings indicated a higher sugar content in smaller particles compared to larger ones in both the APF and CPF. This phenomenon might be attributed to the increased specific surface area of finer particles, enhancing the release of soluble sugars during the extraction process.

Organic acids, characterized by one or more carboxyl functional groups in their structure, occur naturally in plants, animals, and microorganisms [52,53]. As indicated in Table 2, the APF predominantly contained malic acid, ranging from 143.8 to 445.0 mg/100 g dw, while the CPF was rich in lactic acid, ranging from 324.2–430.1 mg/100 g dw. It is interesting to observe that for the APF, the total organic acid increased with lower particle sizes, but the opposite happened for the CPF. This might be explained by the different tissue compositions of apples and carrots: while the coarser APF fractions were mainly constituted by the seeds, peduncles, and peel, the smaller particle size fractions were enriched with the flesh of the fruit. Because the organic acid composition mentioned earlier—malic, acid, ascorbic, acetic, and lactic acids—is primarily found in the flesh of apples, these fractions showed higher organic acid contents [54]. These acids contribute to the overall taste, flavor, and acidity of the fruit when consumed [55]. In the CPF, the opposite occurred, and the fraction with coarser particles showed a higher organic acid content. This happened because these fractions were mainly composed of fleshy tissue, which consists mainly of parenchyma cells and is rich in organic acids [56].

The lower pH observed in the APF (around 4.1) compared to the CPF (around 5.4) can be attributed to the higher acidic capacity of malic and citric acids, which have pKa values of 3.5 and 2.8, respectively, in contrast to lactic and acetic acids, with pKa values of 3.8 and 4.8, respectively [57]. Despite noticeable variations in the organic acid contents among the different fractions of both pomaces, the results did not exhibit any significant correlation between the PS of the pomace and its organic acid content.

### 3.4. Carotenoids

Carotenoids are red- to yellow-colored lipid-soluble pigments that naturally occur in fruits and vegetables and exhibit antioxidant activity by quenching free radicals, minimizing the damages of reactive oxidants and inhibiting lipid peroxidation [58]. Lutein and β-carotene were the most abundant carotenoids in the APF and CPF, ranging from 0.2 to 0.4 mg/100 g dw and from 0.3 to 0.7 mg/100 g dw, respectively (Figure 8). In the APF, while fraction D showed the highest content of lutein, fraction E presented the lowest content of both lutein and β-carotene. Regarding the CPF, there were no significant differences between the different fractions in terms of lutein content, while fraction C was enriched with β-carotene. This trend was observed in both flours for both lutein and β-carotene, which is explained by the fact that a reduction in PS enhances the surface area, potentially increasing carotenoid extraction efficiency and, therefore, the total carotenoid content [59]. Yet, during the drying process of the pomaces, this also heightened the exposure to heat of the carotenoids near the particle surface, causing their degradation and decreasing their amount in smaller particle fractions [60].

### 3.5. Vitamins

The vitamin content results are presented in Table 3. Vitamin E is lipophilic and was only detected in the APF, as the content of lipids in the CPF was negligible. The amount of this vitamin in the APF showed a decreasing trend from 2.0 to 1.5 mg/100 g dw as the size of the particles decreased, reaching a minimum in the smallest particles (fraction E). As mentioned earlier, this phenomenon may stem from the compound’s degradation during air-drying, exacerbated by its increased exposure to air and heat as the PS decreased.

Vitamins B6 and C are water-soluble and range from 0.7 to 1.1 mg/100 g dw and 33.3 to 44.7 mg/100 g dw and from 0.7–0.9 to 7.2–12.6 mg/100 g dw in APF and CPF, respectively. The vitamin C content in the APF was almost four-fold higher than in the CPF. In both flours, the reduction in PS, especially for particles smaller than 1 mm, generally resulted in an increase in vitamins B6 and C. This trend can be attributed to improved extraction yield associated with an increase in surface area. Interestingly, in the case of apple pomace, the highest concentrations of both vitamins were observed in particles larger than 1 mm (fraction B). Thais could be due to the positive impact of the smaller surface area in minimizing the exposure of the nutrients to heat during the drying process [60].

Referring to Table 3, it is evident that APF stands out as a notable source of both vitamin C and B6. In contrast, CPF appears to be a significant source of exclusively vitamin B6.

### 3.6. Total Phenolic Content (TPC) and Antioxidant Activity (AAT)

The TPC of the APF and CPF ranged between 185.9 and 328.7 and 36.2 and 84.9 mg GAE/g dw, respectively (Figure 9). These results are in accordance with the literature for pomaces dried at 80 °C [61]. For both pomaces, the fractions with smaller particles (D and E) had the highest amount of phenolic compounds, while in the largest particles, it was the lowest (93.0 and 18.1 mg GAE/g dw for APF and CPF, respectively). In the finer particles, the increase in surface area and the disruption of the cell wall resulted in a higher yield during the extraction of phenolic compounds [62].

The results for the radical scavenging activity (DPPH) and ferric reducing antioxidant power (FRAP) showed the same trend as the TPC, with the smallest particles (fractions D and E) showing the highest activity in both the APF and CPF (Figure 10).

## 4. Conclusions

Our findings showed that the fractions containing smaller particles (D and E) had significantly higher contents of fat, sugar, phenolic compounds, and antioxidant activity, whereas the fiber content was more pronounced in those with larger particles (B and C). These results demonstrate the importance of PS as an important factor worth considering at the time of pomace flour preparation. In the case of AP, the fractionation process resulted in two main groups: fractions rich in fiber and those rich in sugar and phenolic compounds. These results support the idea of dividing AP based on PS as a promising approach to separate the pomace for different applications in terms of functionality. Additionally, the two fractions previously mentioned were rich in minerals such as Na, K, Ca, Mg, P, and Fe, particularly in the CPF, organic acids including malic acid in the APF, and lactic acid in the CPF. As a result, both fractions show potential to be used as valuable ingredients to further improve food products in terms of nutritional value and sensory properties.

## Figures and Tables

**Figure 1 foods-13-00710-f001:**
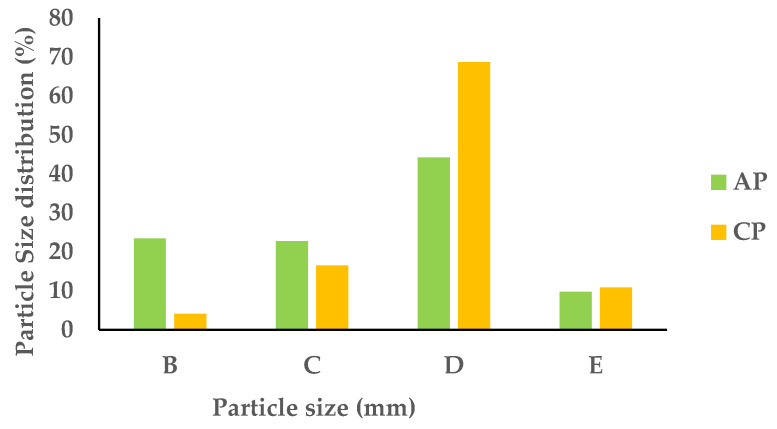
PS distribution between the different granulometric fractions of APF and CPF ((B): PS ≥ 1 mm; (C): 0.71 ≥ PS > 0.18 mm; (D): 0.18 ≥ PS > 0.075 mm; (E): PS < 0.075 mm).

**Figure 2 foods-13-00710-f002:**
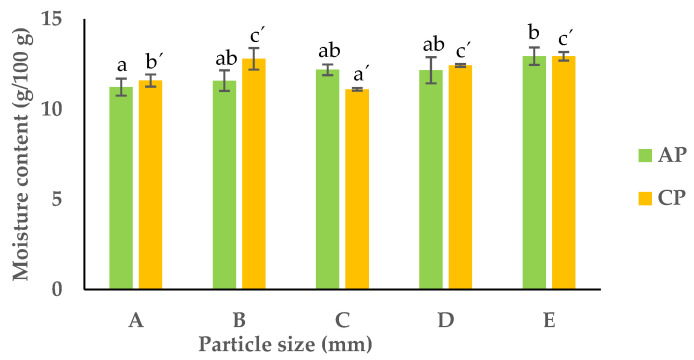
Moisture content of the different fractions of APF and CPF ((A) unfractionated pomace; (B) PS ≥ 1 mm; (C) 0.71 ≥ PS > 0.18 mm; (D) 0.18 ≥ PS > 0.075 mm; (E) PS < 0.075 mm). The results are shown as the mean ± standard deviation of three replicates. Different lowercase letters a–c and a’–c’ indicate significant differences between the different fractions of AP and CP, respectively (*p* < 0.05).

**Figure 3 foods-13-00710-f003:**
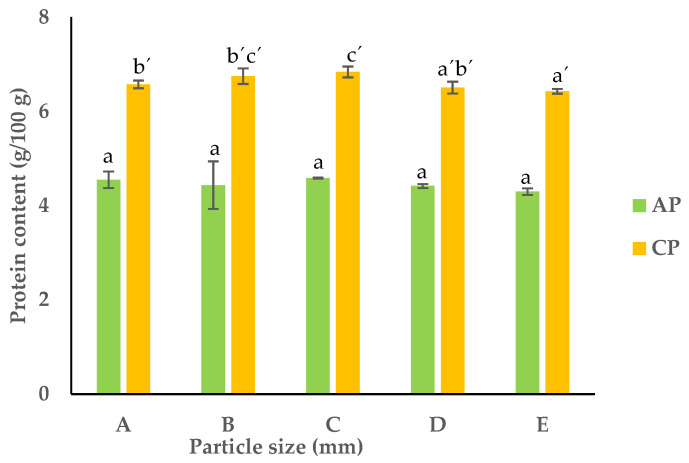
Protein content of APF and CPF ((A) unfractionated pomace; (B) PS ≥ 1 mm; (C) 0.71 ≥ PS > 0.18 mm; (D) 0.18 ≥ PS > 0.075 mm; (E) PS < 0.075 mm). The results are shown as the mean ± standard deviation of three replicates. Different lowercase letters a–c and a’–c’ indicate significant differences between the different fractions of AP and CP, respectively (*p* < 0.05).

**Figure 4 foods-13-00710-f004:**
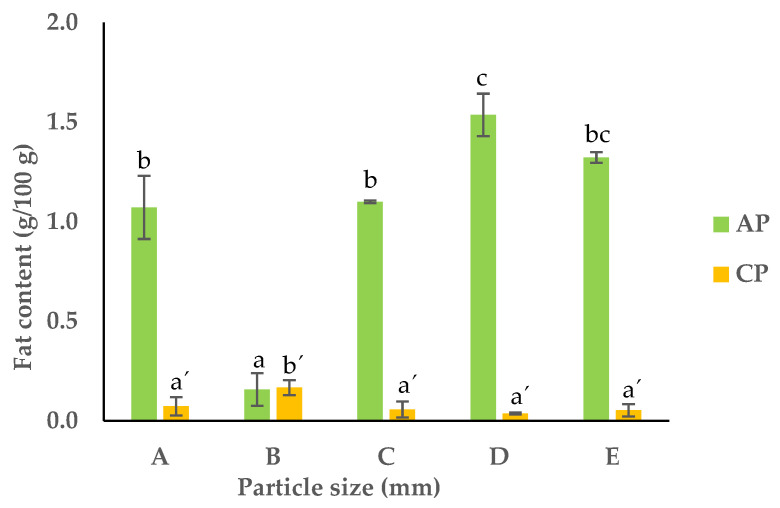
Fat content of APF and CPF ((A) unfractionated pomace; (B) PS ≥ 1 mm; (C) 0.71 ≥ PS > 0.18 mm; (D) 0.18 ≥ PS > 0.075 mm; (E) PS < 0.075 mm). The results are shown as the mean ± standard deviation of three replicates. Different lowercase letters a–c and a’–c’ indicate significant differences between the different fractions of AP and CP, respectively (*p* < 0.05).

**Figure 5 foods-13-00710-f005:**
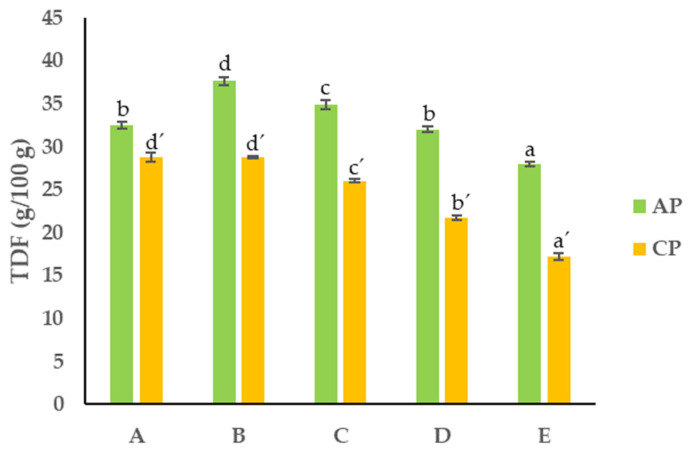
Total dietary fiber content of APF and CPF ((A) unfractionated pomace; (B) PS ≥ 1 mm; (C) 0.71 ≥ PS > 0.18 mm; (D) 0.18 ≥ PS > 0.075 mm; (E) PS < 0.075 mm). The results are shown as the mean ± standard deviation of three replicates. Different lowercase letters a–d and a’–d’ indicate significant differences between the different fractions of AP and CP, respectively (*p* < 0.05).

**Figure 6 foods-13-00710-f006:**
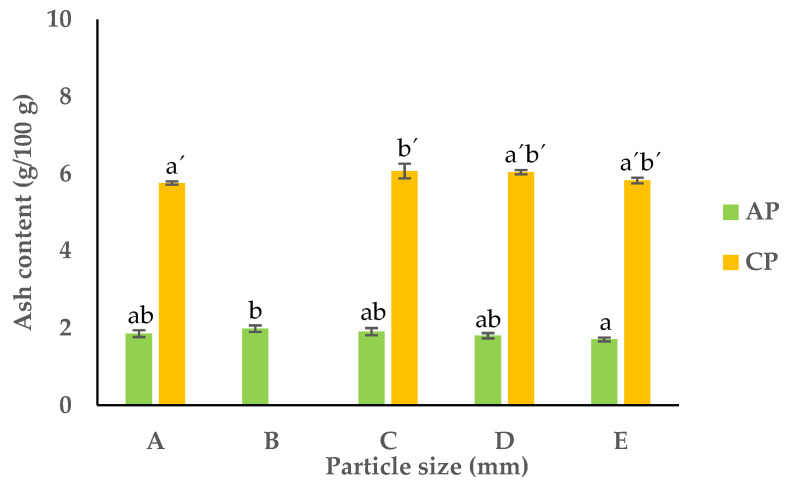
Ash content of APF and CPF ((A) unfractionated pomace; (B) PS ≥ 1 mm; (C) 0.71 ≥ PS > 0.18 mm; (D) 0.18 ≥ PS > 0.075 mm; (E) PS < 0.075 mm). The results are shown as the mean ± standard deviation of three replicates. Different lowercase letters a–c and a’–c’ indicate significant differences between the different fractions of AP and CP, respectively (*p* < 0.05).

**Figure 7 foods-13-00710-f007:**
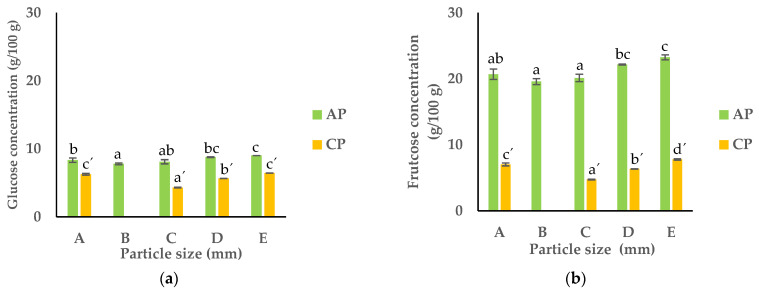
Carbohydrate contents: (**a**) glucose and (**b**) fructose of APF and CPF ((A) unfractionated pomace; (B) PS ≥ 1 mm; (C) 0.71 ≥ PS > 0.18 mm; (D) 0.18 ≥ PS > 0.075 mm; (E) PS < 0.075 mm). The results are shown as the mean ± standard deviation of two replicates. Different lowercase letters a–c and a’–c’ indicate significant differences between the different fractions of AP and CP, respectively (*p* < 0.05).

**Figure 8 foods-13-00710-f008:**
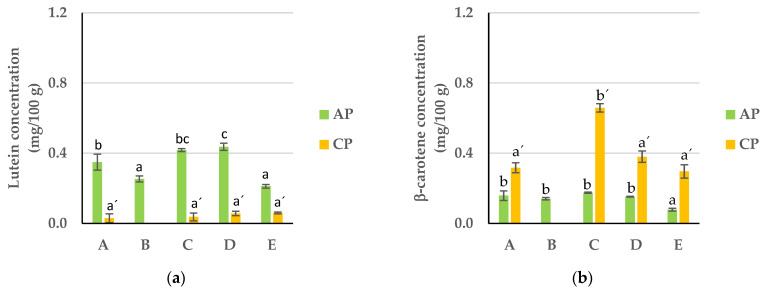
Carotenoid content: (**a**) lutein and (**b**) β-carotene in APF and CPF ((A) unfractionated pomace; (B) PS ≥ 1 mm; (C) 0.71 ≥ PS > 0.18 mm; (D) 0.18 ≥ PS > 0.075 mm; (E) PS < 0.075 mm). The results are shown as the mean ± standard deviation of two replicates. Different lowercase letters a–c and a’–c’ indicate significant differences between the different fractions of AP and CP, respectively (*p* < 0.05).

**Figure 9 foods-13-00710-f009:**
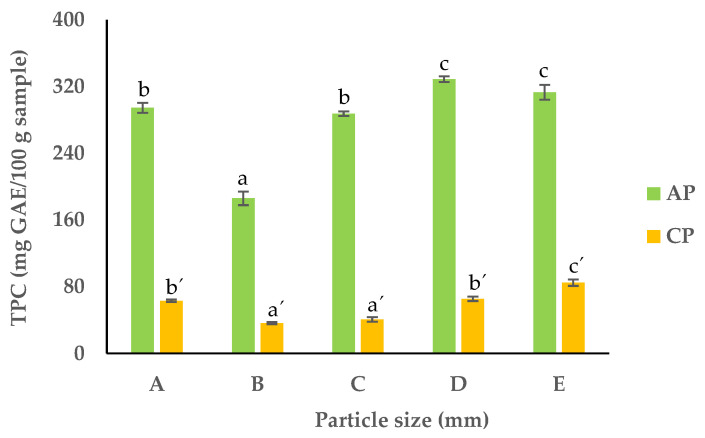
TPC in APF and CPF ((A) unfractionated pomace; (B) PS ≥ 1 mm; (C) 0.71 ≥ PS > 0.18 mm; (D) 0.18 ≥ PS > 0.075 mm; (E) PS < 0.075 mm). The results are shown as the mean ± standard deviation of three replicates. Different lowercase letters a–c and a’–c’ indicate significant differences between the different fractions of AP and CP, respectively (*p* < 0.05).

**Figure 10 foods-13-00710-f010:**
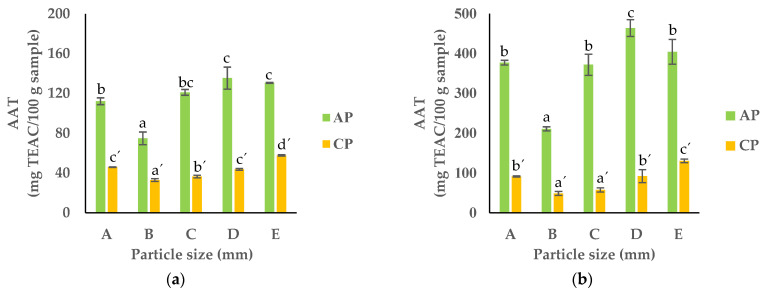
AAT measured based on (**a**) DPPH and (**b**) FRAP of APF and CPF ((A) unfractionated pomace; (B) PS ≥ 1 mm; (C) 0.71 ≥ PS > 0.18 mm; (D) 0.18 ≥ PS > 0.075 mm; (E) PS < 0.075 mm). The results are shown as the mean ± standard deviation of three replicates. Different lowercase letters a–c and a’–c’ indicate significant differences between the different fractions of AP and CP, respectively (*p* < 0.05).

**Table 1 foods-13-00710-t001:** Mineral contents of APF and CPF. The results are the mean of triplicates, shown as the mean ± standard deviation, and are reported as mg of mineral per 100 g of pomace flour.

Sample	Fractions	Na	K	Ca	Mg	P	S	Fe	Cu	Zn	Mn	B
AP	A	50.1 ± 5.18 ^b^	917.8 ± 2.29 ^ab^	65.9 ± 3.88 ^b^	61.6 ± 1.34 ^b^	105.9 ± 5.48 ^b^	53.9 ± 2.49 ^b^	19.9 ± 8.53 ^a^	0.8 ± 0.08 ^bc^	1.7 ± 1.13 ^a^	1.6 ± 0.15 ^ab^	4.9 ± 0.20 ^a^
	B	43.5 ± 2.94 ^ab^	911.9 ± 18.13 ^ab^	66.3 ± 2.51 ^b^	60.9 ± 2.66 ^b^	112.0 ± 9.12 ^b^	54.2 ± 3.04 ^b^	31.8 ± 1.15 ^a^	0.6 ± 0.01 ^a^	0.9 ± 0.13 ^a^	1.3 ± 0.04 ^a^	4.5 ± 0.12 ^a^
	C	41.7 ± 1.54 ^a^	925.3 ± 11.54 ^b^	63.6 ± 3.80 ^ab^	61.6 ± 1.31 ^b^	108.9 ± 3.28 ^b^	50.1 ± 1.63 ^ab^	5.9 ± 3.55 ^a^	0.7 ± 0.03 ^ab^	0.9 ± 0.14 ^a^	1.5 ± 0.02 ^ab^	4.7 ± 0.08 ^a^
	D	39.5 ± 0.87 ^a^	907.9 ± 34.37 ^ab^	64.3 ± 2.65 ^ab^	61.6 ± 0.75 ^b^	104.6 ± 1.86 ^b^	50.1 ± 1.06 ^ab^	86.7 ± 20.02 ^b^	0.9 ± 0.07 ^c^	2.5 ± 1.33 ^a^	1.8 ± 0.17 ^b^	4.9 ± 0.30 ^a^
	E	39.3 ± 0.58 ^a^	872.8 ± 15.01 ^a^	57.0 ± 2.36 ^a^	51.5 ± 1.36 ^a^	90.6 ± 1.51 ^a^	45.7 ± 0.68 ^a^	46.5 ± 8.76 ^ab^	0.8 ± 0.01 ^abc^	1.2 ± 0.24 ^a^	1.5 ± 0.20 ^ab^	4.5 ± 0.09 ^a^
CP	A	347.4 ± 8.11 ^b^	2363.1 ± 63.48 ^ab^	318.0 ± 5.53 ^b^	123.3 ± 2.46 ^b^	239.4 ± 6.99 ^a^	98.0 ± 2.40 ^a^	67.6 ± 8.39 ^b^	0.6 ± 0.01 ^a^	2.3 ± 0.28 ^a^	1.7 ± 0.05 ^ab^	2.6 ± 0.04 ^b^
	B	367.2 ± 1.97 ^c^	2468.5 ± 48.47 ^c^	280.2 ± 10.35 ^a^	118.3 ± 3.76 ^ab^	248.1 ± 8.26 ^a^	103.9 ± 2.03 ^b^	60.0 ± 17.66 ^b^	0.6 ± 0.01 ^ab^	2.2 ± 0.23 ^a^	1.5 ± 0.14 ^a^	2.5 ± 0.08 ^a^
	C	355.6 ± 7.96 ^bc^	2371.0 ± 15.49 ^abc^	291.3 ± 14.58 ^a^	115.8 ± 1.09 ^a^	241.3 ± 2.84 ^a^	97.8 ± 2.11 ^a^	8.5 ± 1.66 ^a^	0.6 ± 0.02 ^a^	1.9 ± 0.17 ^a^	1.5 ± 0.05 ^a^	2.4 ± 0.05 ^a^
	D	348.5 ± 1.78 ^b^	2432.5 ± 25.59 ^bc^	346.3 ± 0.81 ^c^	131.9 ± 0.73 ^c^	249.1 ± 1.55 ^a^	100.7 ± 0.83 ^ab^	7.5 ± 0.82 ^a^	0.7 ± 0.08 ^b^	2.1 ± 0.02 ^a^	1.8 ± 0.01 ^bc^	2.8 ± 0.04 ^c^
	E	328.2 ± 2.77 ^a^	2269.8 ± 18.97 ^a^	359.0 ± 3.25 ^c^	132.4 ± 1.42 ^c^	238.1 ± 1.83 ^a^	96.6 ± 0.50 ^a^	9.8 ± 0.06 ^a^	0.6 ± 0.01 ^ab^	2.1 ± 0.02 ^a^	1.9 ± 0.02 ^c^	2.7 ± 0.04 ^bc^
DRV * (mg/day)	Male	NM	3500	950–1000	350	550	NM	11	1.6	9.4–16.3	3	NM
	Female				300			11–16	1.3	7.5–12.7		

* Dietary reference values for nutrients according to a technical report by the European Food Safety Authority (EFSA) [50]. NM, not mentioned. Superscript, lowercase letters indicate significant differences between the different fractions (*p* < 0.05).

**Table 2 foods-13-00710-t002:** Organic acid (mg/100 g dw) content of APF and CPF. The results are the mean of duplicates shown as the mean ± standard deviation.

Sample	Fraction	Acetic Acid	Tartaric Acid	Lactic Acid	Malic Acid	Ascorbic Acid	Citric Acid	Total Organic Acid
AP	A	44.7 ± 7.16 ^a^	nd	43.8 ± 0.45 ^b^	143.8 ± 1.67 ^a^	33.3 ± 3.46 ^a^	28.8 ± 0.30 ^c^	294.4
	B	36.9 ± 2.75 ^a^	nd	43.4 ± 1.69 ^b^	199.9 ± 12.07 ^b^	44.7 ± 0.16 ^b^	27.5 ± 3.39 ^bc^	352.4
	C	37.4 ± 7.47 ^a^	nd	65.4 ± 1.69 ^c^	233.8 ± 13.16 ^c^	34.6 ± 0.86 ^a^	16.3 ± 0.07 ^ab^	387.5
	D	31.3 ± 7.95 ^a^	nd	69.3 ± 2.61 ^c^	217.3 ± 6.27 ^bc^	37.3 ± 2.19 ^ab^	13.6 ± 3.98 ^a^	368.8
	E	27.9 ± 0.10 ^a^	nd	32.5 ± 2.00 ^a^	445.0 ± 0.58 ^d^	41.6 ± 1.88 ^ab^	28.7 ± 5.24 ^bc^	575.7
CP	A	84.5 ± 6.89 ^b^	25.3 ± 19.82 ^a^	324.2 ± 24.10 ^a^	34.6 ± 0.62 ^a^	7.4 ± 0.23 ^a^	nd	476.0
	C	62.6 ± 2.86 ^a^	15.5 ± 1.89 ^a^	430.1 ± 7.15 ^b^	37.7 ± 4.56 ^a^	7.2 ± 0.42 ^a^	nd	553.1
	D	77.7 ± 6.13 ^ab^	18.0 ± 0.85 ^a^	368.3 ± 16.90 ^ab^	57.6 ± 2.75 ^b^	9.9 ± 0.16 ^b^	nd	531.5
	E	77.1 ± 0.57 ^ab^	13.2 ± 1.60 ^a^	377.3 ± 7.39 ^ab^	57.0 ± 2.66 ^b^	12.7 ± 0.27 ^c^	nd	537.3

Superscript, lowercase letters indicate significant differences between the different fractions of each pomace type *(p* < 0.05).

**Table 3 foods-13-00710-t003:** Vitamin contents (mg/100 g dw) of APF and CPF. The results are the mean of triplicates, shown as the mean ± standard deviation.

Sample	Fraction	Vitamin E	Vitamin B6	Vitamin C
AP	A	1.74 ± 0.057 ^ab^	0.86 ± 0.085 ^a^	33.28 ± 3.460 ^a^
	B	1.97 ± 0.129 ^b^	1.13 ± 0.049 ^b^	44.69 ± 0.163 ^c^
	C	1.88 ± 0.262 ^ab^	0.86 ± 0.014 ^a^	34.56 ± 0.862 ^ab^
	D	1.72 ± 0.021 ^ab^	0.90 ± 0.021 ^ab^	37.34 ± 2.187 ^b^
	E	1.47 ± 0.014 ^a^	0.74 ± 0.071 ^a^	41.56 ± 1.884 ^bc^
CP	A	nd	0.77 ± 0.030 ^a^	7.42 ± 0.232 ^a^
	C	nd	0.72 ± 0.021 ^a^	7.19 ± 0.421 ^a^
	D	nd	0.78 ± 0.047 ^a^	9.90 ± 0.159 ^b^
	E	nd	0.85 ± 0.078 ^a^	12.65 ± 0.273 ^c^
DRV * (mg/day)	Men	13	1.7	110
	Women	11	1.6	95

* Dietary reference values (DRV) for nutrients according to a technical report by the European Food Safety Authority (EFSA) [50]. Superscript, lowercase letters indicate significant differences between the different fractions of each pomace type (*p* < 0.05). nd: not detected.

## Data Availability

The original contributions presented in the study are included in the article, further inquiries can be directed to the corresponding author.

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
