# Peer review of "Effects of Particle Size on Physicochemical and Nutritional Properties and Antioxidant Activity of Apple and Carrot Pomaces"

_foods, 2024, doi:10.3390/foods13050710_

Round 1

Reviewer 1 Report

Comments and Suggestions for Authors

Title: Effect of particle size on physicochemical and nutritional properties and antioxidant activity of apple and carrot pomaces

Manuscript No.: Foods-2857684

This work was dedicated to analyze apple and carrot pomace from their respective industries to find their physicochemical and nutritional properties as well as antioxidant activity as affected by the powder particle size. This is an interesting work holding a practical application from various aspect of science and technology. The manuscript is well written, while the materials and methods along with the results are also well illustrated using clear figures and tables. However, I have a single major concern and few minor suggestions.

Major concern: Smaller particles size resulted higher content of fat, sugar, phenolic compounds and antioxidant activity, while lower content of fiber (Conclusion; line 438-440), while mineral content and organic acid content were also different with particle size (conclusion; line 445-447). As the particle size reduction is totally physical process, how it impacted these chemical properties (contents of fat, sugar, phenolic compounds, fibers, minerals, and organic acids)? This concern is for all the results and discussion sections. Please give the reasons behind this phenomenon.

Minor concerns:

·        Line 59: First define PS, then use abbreviation.

·        Line 3220: Please use 3220 ×g, instead of 3220 g.

·   Line 151 and 175: Please modify “by [27]” and “by [28]” appropriately.

·   Please use mL and µL, instead of ml and µl, respectively (throughout the manuscript).

·        “p < 0.05”: p should be italicized.

·        Section 3.1: Please modify the sub-title appropriately.

·      Line 205 (Not correct, it should be D: 0.18 ≥ PS > 0.075 mm) and line 207 (Not correct; it should be E: < 0.075 mm)?

·     Fig. 1 to Fig. 9: please write the meaning of lower-case a, b, c with its p value.

·     Fig. 9 for two figures (Two Fig. 9) – please amend it accordingly.

Comments on the Quality of English Language

English language is OK.

Reviewer 2 Report

Comments and Suggestions for Authors

The manuscript "Effect of particle size on physicochemical and nutritional properties and antioxidant activity of apple and carrot pomaces" is a good work of characterización of different particle size flours sieved from dried wastes from the juice industry of pomace and carrot.

The chemical composition should be improved by measuring Total Dietary Fiber by the enzymogravimetric assay. The nutritional profile was developed according the the common analysis usually performed to flours: minerals, antioxidants, etc.

I suggeted some revisions and included suggestions in the pdf version of the manuscript.

Comments on the Quality of English Language

The quality of English language is acceptable.

Round 2

Reviewer 2 Report

Comments and Suggestions for Authors

This manuscript was deeply improved. Suggestions of all reviewers were included in the text, figures and legends. Bibliograpy was corrected according reviewer's suggestions.